# Lexical competition in the flankers task revisited

**Aaron Vandendaele** [1] *, **Jonathan Grainger** [2,3]

**1** Department of Experimental Psychology, Ghent University, Ghent, Belgium, **2** Laboratoire de Psychologie Cognitive, CNRS & Aix-Marseille University, Marseille, France, **3** Institute of Language, Communication and the Brain, Aix-Marseille University, Marseille, France

* aaron.vandendaele@ugent.be

## Abstract

We investigated the impact of flanking stimuli that are orthographic neighbors of central target words in the reading version of the flankers task. Experiment 1 provided a replication of the finding that flanking words that are orthographic neighbors of central target words (e.g., BLUE BLUR BLUE) facilitate lexical decisions relative to unrelated word flankers (e.g., STEP BLUR STEP). Experiment 2 tested the hypothesis that this facilitatory effect might be due to the task that was used in Experiment 1 and in prior research–the lexical decision task. In Experiment 2 the task was perceptual identification, and here we observed that orthographic neighbor flankers interfered with target word identification. Experiment 2 also included a bigram flanker condition (e.g., BL BLUR UE), and here the related bigram flankers facilitated target word identification. We conclude that when the task requires identification of a specific word, effects of lexical competition emerge over and above the facilitatory effects driven by the sublexical spatial pooling of orthographic information across target and flankers, and that the inhibitory influence of lexical competition has an even stronger impact when flankers are whole words.

## Introduction

It is a well-established fact that single word recognition is influenced by a word's orthographic relatedness with other words, such as documented by the numerous studies investigating the effects of orthographic neighborhood (e.g., Andrews [1]; Grainger [2]). A number of studies have also investigated how orthographic relatedness affects word recognition during sentence reading, both in terms of the orthographic neighborhood of the individual words in the sentence (e.g., Pollatsek et al. [3]), and by manipulating the orthographic relatedness of parafoveal previews that function as a prime stimulus prior to readers fixating a given target word at the same location as the preview (Williams et al. [4]). In the present study we focus on effects of orthographic relatedness across words that appear at different locations in a sequence of words, and more specifically the effects of orthographic relatedness across adjacent words. This is important because it addresses the key issue of the extent to which more than one word can be processed in parallel during sentence reading, and how such parallel processing can influence sentence reading (see Snell & Grainger [5], for a review).

**Data Availability Statement:** All materials, data and analyses scrips can be accessed through following link: https://osf.io/8fwem.

**Funding:** 'This project was funded by European Research Council (ERC) grant 742141 awarded to Jonathan Grainger and by Flanders Research

Foundation (FWO) grant 1154021N awarded to Aaron Vandendaele. The funders had no role in study design, data collection and analysis, decision to publish, or preparation of the manuscript'.

**Competing interests:** The authors have declared that no competing interests exist.

Effects of orthographic relatedness across adjacent words during sentence reading have been investigated using a parafoveal-on-foveal manipulation, such that when fixating word N, the word immediately to the right (N+1) can be orthographically related to word N or not, and as readers' gaze moves to position N+1 the word at that location is changed to become a regular continuation of the sentence (e.g., "The slight blur *blue* the shape of . . ." = > "The slight blur *took* the shape of . . .", where the target is the word "blur", and italics indicate the stimulus change at N+1). Orthographic relatedness has been found to facilitate processing of the target word (i.e., shorter gaze durations) when the parafoveal stimulus (N+1) is both a word and a nonword (Angele et al. [6]; Dare & Shillcock [7]; Inhoff et al. [8]; Mirault et al. [9]; Snell et al. [10]).

In the present study we provide a further examination of the impact of orthographic relatedness across adjacent words within a sequence of words presented simultaneously and aligned horizontally. To do so we use a simplified reading paradigm introduced by Dare and Shillcock [7]. In their reading version of the flankers task, Dare and Shillcock presented participants with a central target word that was flanked to the left and to the right by letters that were related or not to the target (e.g., RO ROCK CK vs. BA ROCK TH). Participants were instructed to perform a lexical decision on the central targets and could ignore the flanking letters. Orthographically related flankers were found to facilitate lexical decisions to central targets (see also Cauchi et al. [11]; Grainger et al. [12]; Snell, Bertrand et al. [13]). Crucially, with respect to the goals of the present study, these facilitatory flanker effects are also found when the related flankers form an orthographic neighbor of the target (e.g., BL BLUR UE, with the flankers BL and UE combining to form the word "blue": Snell et al. [10]). Even more relevant with respect to the present work is that the facilitatory effects of orthographically related flankers are also observed when the flankers are full words (e.g., BLUE BLUR BLUE: Snell, Bertrand, Meeter, et al. [14]).

Grainger et al. [12] proposed that the facilitatory effects of orthographically related parafoveal stimuli are driven by the spatial pooling of orthographic information spanning multiple spatially distinct stimuli into a single processing channel. The relative positions of letters present in the foveal and parafoveal stimuli are encoded in this central processing channel, and these then activate location-invariant orthographic representations of words (see Fig 1). Orthographically related parafoveal stimuli therefore contribute to foveal target word activation, hence the observed facilitation. The fact that facilitation is still obtained when the flanking stimuli are whole words, as in Snell, Bertrand, Meeter et al. [14], raises the question as to existence of lateral inhibitory connections between co-active lexical representations in the central processing channel, as postulated in the OB1-reader model of word recognition and text reading (Snell, van Leipsig, et al. [15]).

However, these facilitatory effects of orthographically related parafoveal words contrast with the interference found by Mirault et al. [16] in a grammatical decision experiment with word sequences containing adjacent orthographically related words (e.g., The *brave brace* the wind vs. The *brave daunt* the wind). In the present study we test one possible interpretation of the contrasting findings of prior studies investigating effects of adjacent orthographically related stimuli during multi-word processing (i.e., in sentence reading and the reading version of the flankers task). The backbone of this account is a trade-off between the facilitation driven by the spatial pooling of orthographic information across adjacent stimuli when they are related, and the inhibition driven by the co-activation of lexical representations when the task involves unique word identification. Such inhibitory influences are greatly reduced or even absent when the task does not necessarily require unique word identification, such as the lexical decision task (e.g., Grainger & Jacobs [17]), and the decision to move the eyes to the next word in a sentence (e.g., Reichle et al., [18, 19], however, see Pollatsek et al. [3] for inhibitory effects of orthographic neighborhood size in gaze durations with the same stimuli that showed

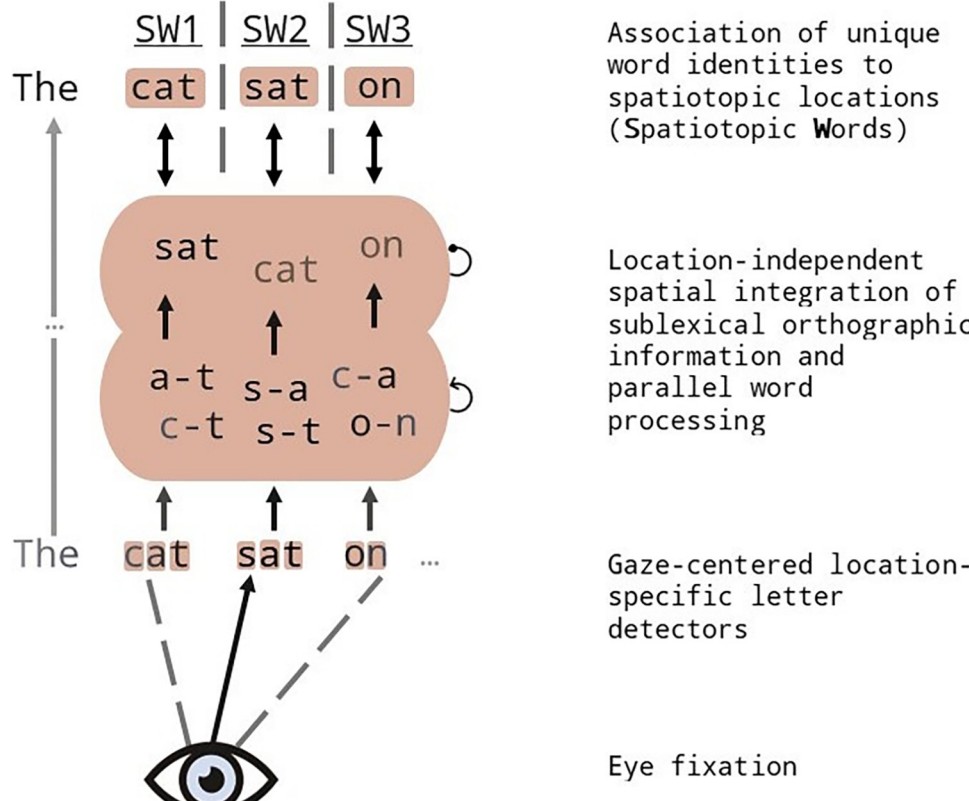

**Fig 1. Illustration of the spatial integration of orthographic information proposed by Grainger et al. [12] and later incorporated in the architecture of OB1-reader (Snell, van Leipsig, et al. [15]).** Information about letter identities and their locations is pooled across multiple words into a single channel for location-invariant sublexical orthographic processing (via a bag-of-bigrams) and parallel word processing (bag-of-words). Relative activation levels of co-active words (illustrated by bold font) are determined by acuity, crowding, spatial attention, and length-matching. Co-active word representations compete for identification via lateral inhibitory connections. Note that this is a simplified version of the complete model that neither includes bigrams formed using interword space information (e.g., T#, #S), nor a bag of position-independent letters that would provide input to the competing word "in" (see [13]).

facilitation in a lexical decision task). Therefore, when measuring lexical decision responses or gaze durations (Snell et al. [10]), facilitatory effects of spatial pooling dominate. On the other hand, when the task requires unique word identification, as is necessary in order to make an accurate grammatical decision (Mirault et al. [16]), then inhibitory effects of lexical competition dominate.

The present study puts this explanation to test by manipulating i) the task performed by participants (lexical decision in Experiment 1 and perceptual identification in Experiment 2), and ii) the nature of the flanking stimuli (whole words or bigrams in Experiment 2). In Experiment 1 we start by replicating the lexical decision experiment of Snell et al. [10] using a different set of stimuli (i.e., French instead of Dutch, as used in the original study). Here, we expected to observe facilitatory effects of orthographically related flanker words (BLUE BLUR BLUE) hence replicating Snell et al. [10]. In Experiment 2 the task was changed to perceptual identification, where participants had to identify central target words. We used the same set of target and flanker words as in Experiment 1 while introducing a further manipulation: flankers could either be whole words (BLUE BLUR BLUE) as in Experiment 1, or bigrams (BL BLUR UE). We predicted that inhibitory effects of flanker relatedness should be found when the task involves word identification, and more so with word flankers than bigram flankers.

## Experiment 1: Lexical decision

### Methods

**Participants.** Ninety participants (45 male, mean age = 27.24, *SD* = 6.01, min = 18, max = 40) took part in this online experiment. All participants indicated being native speakers of French without dyslexia or other neurological disorders. Participants were recruited using the Prolific platform (Palan & Schitter [20]) and paid at the rate of £9/hour.

**Materials and design.** From the French Lexicon Project (Ferrand et al. [21]), we first selected 90 four-letter French target words with an average LDT of 689 ms and a frequency of 3.79 Zipf (van Heuven et al. [22]). Following Snell et al. [10], each target word was then paired with an orthographically related flanker and a control flanker that was not related to the target. Orthographically related flankers were single substitution neighbors of the target (e.g., BLUE-BLU**R**) with the substitution occurring at all positions. Compared to the target word, both types of flanker had a higher frequency (4.82 & 4.87 Zipf) and had an average LDT (extracted from the French Lexicon Project) at least 40 ms shorter (643 & 649 ms respectively). For each target word we selected a pseudoword target that was matched on length and syllable structure using the Wuggy pseudoword generator (Keuleers & Brysbaert [23]). Targets and flankers did not contain any diacritics. Pseudoword targets were flanked by orthographically related and unrelated words in the same way as word targets. Pseudoword targets were included for the purpose of the lexical decision task, and the data concerning these targets were not analyzed.

**Apparatus and software.** Stimuli and experimental design were implemented using the OpenSesame software (Mathôt et al. [24]) and imported online through the JATOS application (Lange et al. [25]). Participants were instructed to use their personal computer and sit 50 cm from their screen so that each character space subtended approximately 0.53 degrees of visual angle.

**Ethics statement.** Informed consent was obtained from participants online, who checked a box indicating their agreement before proceeding to the experiment. All experiments in this study were performed in accordance with the provisions of the World Medical Association Declaration of Helsinki and ethics approval was obtained from the ethical committee of the faculty of psychology and educational sciences at Ghent University.

**Procedure.** Before the experiment, participants received on-screen instructions in function of which version of the task they were assigned to, together with ten practice trials to get used to the procedure. Fig 2 provides a description of the procedure. Each trial began with two vertically aligned fixation bars that stayed on-screen for 500 ms. Afterwards, a central word embedded between two flankers would appear for a duration of 50 ms. This very short stimulus exposure (much shorter than the typical 150–200 ms used in prior flanker studies) was used in anticipation of Experiment 2. Target and flanker were immediately followed by a post-mark consisting out of hashmarks (####) covering all stimuli. The hashmarks stayed on-screen until participants responded by indicating as rapidly and as accurately as possible whether the central target was a word or not (lexical decision). Participants had a maximum of 2000 ms to respond before feedback was provided in the form of a green (correct) or red (incorrect) circle that stayed on-screen for a duration of 500 to 700 ms, after which a new trial would begin. On average, the experiment lasted about 10 minutes. In total, we obtained 2025 observations per condition (before data exclusion), exceeding the recommended 1600 trials by Brysbaert and Stevens [26] for sufficient statistical power.

### Results

We analyzed response times (RTs) and error rates for the word targets. First of all, we removed the data of 3 participants because their accuracy rates did not reach 50%. Furthermore, the RT

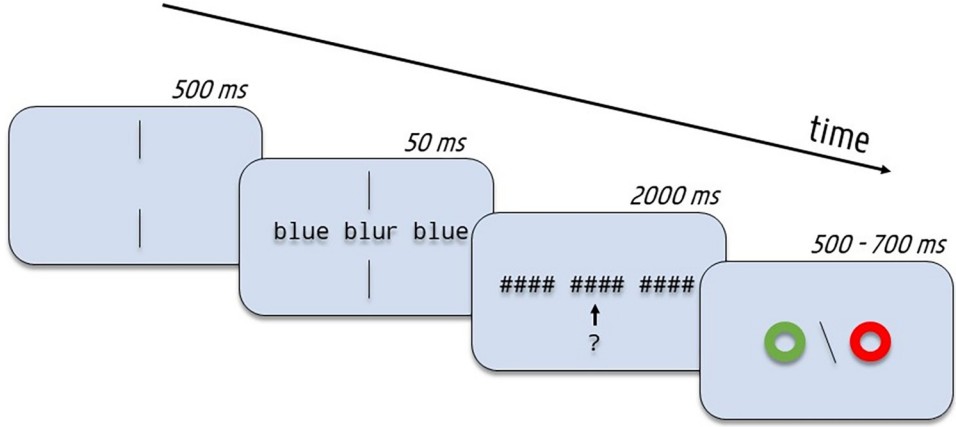

**Fig 2. Example of the trial procedure used in the experiments.** The question mark refers to the task that participants had to perform on the central target stimulus: lexical decision in Experiment 1 and perceptual identification in Experiment 2. It was not part of the actual procedure. Compared with the conditions that were tested in the experiment, the example stimuli are enlarged relative to screen size and are shown here in English instead of French for illustration purposes.

analysis only included trials with correct responses, leading to the exclusion of 23.87% of the data. Lastly, trials which exceeded the 2.5 SD interval from the grand mean were also excluded (2.55%). This resulted in a total number of observations of 1601 for the related flanker condition and 1474 for the control flanker condition. In order to meet the model's assumption that the data are distributed normally, a logarithmic transformation (Log10(RT)) was performed prior to the analyses. We used linear mixed models to analyze RTs and generalized linear mixed models to analyze the error rates. All models were fitted with the lmer and glmer functions from the lme4 package version 1.1–31 (Bates et al. [27]) using R version 4.2.1 statistical computing environment (R Core Team [28]). We started from the most complicated model and reduced complexity until convergence was reached. In the final model, flanker type (orthographic neighbor vs. control) was a fixed effect, items and participants were included as random effects. The RT model allowed for a by-item random slope, whilst the error rates model included both the by-item and by-participant random slopes (Baayen et al. [29]; Barr et al. [30]). We report $b$-values, standard errors (SEs) and $t$- or $z$-values, with those beyond |1.96| deemed as significant. Significant effects are indicated in bold.

There was a main effect of flanker relatedness in RTs ($b$ = 1.98, $SE$ = 0.78, $t$ = 2.54), with faster responses (635 ms) when flankers were orthographically related to targets compared with unrelated flankers (646 ms). The effect of flanker relatedness was also significant in error rates ($b$ = 0.53, $SE$ = 0.14, $z$ = **3.79**), with fewer errors being made to targets in the presence of related flankers (15.2%) compared with unrelated flankers (22.4%).

## Discussion

In Experiment 1, we successfully replicated the findings of Snell, Bertrand et al. [13] while using a much shorter stimulus exposure duration (50 ms compared with 150 ms). Orthographically related flanker words significantly facilitated lexical decisions to central target words. We now turn to the crucial experiment of the present study where we change the task performed on central target words. In Experiment 2 participants had to identify central target words (no pseudoword targets were presented in this experiment) embedded between two flanker stimuli that could either be words (e.g., BLUE BLUR BLUE) as in Experiment 1, or bigrams (e.g., BL

BLUR UE), and were either related or not to the target. They entered their response using the computer keyboard. Apart from the change in task, the procedure was the same as in Experiment 1 (see Fig 2).

## Experiment 2: Perceptual identification

### Methods

**Participants.** Ninety-nine participants (52 male and 1 participant who chose the option "other/undefined", mean age = 26.84, *SD* = 5.99, min = 18, max = 44) took part in this online experiment. All participants indicated being a French native speaker without dyslexia or other neurological disorders. Participants were recruited using the Prolific platform (Palan & Schitter [20]), and paid at the rate of £9/hour. Participants that already performed in the first experiment were ineligible to participate.

**Materials and design.** The materials and design were similar to Experiment 1 with the exception that there were no pseudoword targets, and there were two types of flanker stimuli: bigrams (e.g., BL BLUR UE) or words (e.g., BLUE BLUR BLUE). The word flankers and targets were the same as tested in Experiment 1, and the bigram flankers were derived from the word flankers and tested in a different sub-experiment. The bigram flankers were formed by dividing the word flankers into their initial and final bigrams which were then placed respectively left and right of the target and separated by a single space. Participants were either be assigned to the bigram flanker version (n = 48) or the word flanker version (n = 51). Due to the online testing procedure, counterbalancing based on a linearly increasing variable such as participant number was not possible. As suggested by Mathôt and March [31], participants were randomly assigned to the bigram version or flanker version of the task, resulting in a slight imbalance in participant numbers per type of flanker. Within each sub-experiment, orthographic relatedness between target and flanker words was manipulated using a Latin-square design, such that each target word was seen with both orthographically related and unrelated flankers, but only once per participant.

**Apparatus, software, and procedure.** These were the same as in Experiment 1, with the exception that participants now had to identify and type the target word they perceived using their keyboard. Instead of a maximum response time, participants now had unlimited time to type their response. Afterwards, feedback was provided in the form of a green (correct) or red (incorrect) circle that stayed on-screen for a random duration between 500 and 700 ms, after which a new trial would begin. In total, we obtained 2295 (word flankers) and 2160 (bigram flankers) observations per flanker relatedness condition.

### Results

The same GLME analysis as in Experiment 1 was performed on error rates, with an added factor that indicated the type of flanker (bigrams or flankers). The model structure did not allow for any random slopes. We observed a main effect of flanker relatedness (b = 0.53, SE = 0.11, z = 4.63), indicating that there were more errors made when flankers were unrelated. There was also a main effect of type of flanker (b = 0.56, SE = 0.23, z = 2.45), meaning that participants made more errors when flankers where bigrams. Crucially, we also observed a significant interaction between flanker relatedness and type of flanker (b = -0.80, SE = 0.15, z = -5.34), indicating that participants made more errors when flankers were related, but only in when the flankers were words (see Fig 3). Follow-up analyses revealed that there was a significant inhibitory effect of flanker relatedness with word flankers (b = -0.28, SE = 0.10, z = -2.70) and a significant facilitatory effect of flanker relatedness with bigram flankers (b = 0.54, SE = 0.12, z = 4.35).

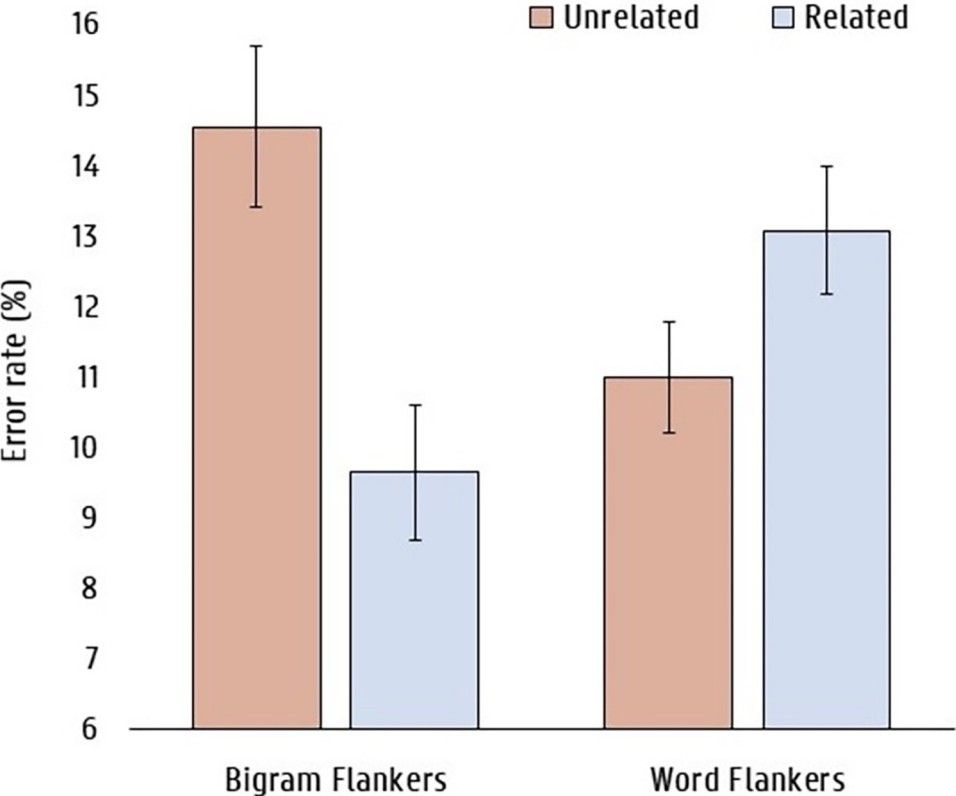

**Fig 3. Mean error rates (in %) as a function of flanker relatedness (unrelated vs. related) and flanker type (bigram vs. word) in Experiment 2.** Error bars indicate 95% confidence intervals.

**Post-hoc analyses.** In these analyses we examined the nature of errors made in Experiment 2. We calculated the percentage of all errors relative to the total number of errors pooled across the related and unrelated flanker conditions. This was done separately for the word flanker and bigram flanker conditions. Error percentages were calculated for the following categories separately for the related and unrelated flanker conditions. The first category of errors were those where *no response* was given. Then we classified all errors where a response was provided as follows. A second category of errors were those that corresponded to the flanker word itself (or the combination of the flankers in the bigram condition–e.g., BL BLUR UE– response "blue"). These were classified as *repetition errors*. Next, non-identical orthographically *related errors* were defined as follows: 1) errors that were orthographic neighbors of flanker stimuli (defined as sharing 3 out of 4 letters at the same position with flanker stimuli, with positions marked from left to right for the bigram flankers–i.e., 12 ABCD 34); 2) errors that were orthographic neighbors of the target; and 3) migration errors, where the erroneous response combined letters from both target and flanker stimuli and contained at least 3 letters at the same position either in the flanker or the target (see Vandendaele et al., [32], for a similar analysis of migration errors). Lastly, all other erroneous responses were classified as *unrelated errors*. The same GLME analysis as in the main analysis was then performed for each category of error, separately for the word and bigram flanker conditions with flanker relatedness as sole fixed factor. The condition means per error category in the related flanker and unrelated flanker conditions are shown in Table 1 for the word flanker condition and Table 2 for the bigram flanker condition, accompanied by the respective *b*-, *SE* and *z*-values for the effect of

**Table 1. Proportion of different errors made per error category in the word flanker condition calculated separately for the related and unrelated flanker conditions.**

| Error Type | Flanker condition | | effect | Relatedness Effect | | |
| | Related | Unrelated | | b | SE | z |
|---|---|---|---|---|---|---|
| No response | 0.4% | 0.3% | -0.1% | -0.27 | 0.49 | -0.55 |
| Repetition error | 4.9% | 0.8% | **-4.1%** | -1.64 | 0.27 | **-6.06** |
| Related error | 4.7% | 5.9% | 1.2% | 0.26 | 0.15 | 1.75 |
| Unrelated error | 1.8% | 2.6% | 1.2% | 0.42 | 0.22 | 1.88 |
| Total | 11.8% | 9.6% | **-2.2%** | -0.28 | 0.10 | **-2.70** |

*Note.* The total error rate corresponds to the data used in the main analysis.

flanker relatedness. Lastly, the number of observations per error condition were limited, we also fitted Bayesian mixed-effects regression modeling using the brms package (Bürkner [33]). All models were fit with 3000 iterations for warm-up and 17000 iterations for sampling. Similar to the frequentist models, flanker relatedness was the sole included predictor. For each error category, we report point & error estimations, the 95% credible interval, the Rhat convergence statistic and the number of effective sample size (ESS). Evidence for an effect was deemed meaningful if the 95% credible interval of the posterior distribution did not include 0. All effects were equivalent in both approaches. Results for these can be found in the S1 Appendix.

**Word flankers.** The results of the error type analysis for word flankers are shown in Table 1. The most striking aspect of this analysis is that the inhibitory effect of flanker relatedness seen in the main analysis is mostly driven by errors that are identical to the flanker stimulus (i.e., *repetition errors*). Indeed, there were actually fewer *related errors* and *unrelated errors* (see definition provided above for the different types of error) in the related flanker condition, although these differences did not reach statistical significance.

**Bigram flankers.** The results of the analysis of error types in the bigram flanker version of the experiment are shown in Table 2. Here we see that the facilitatory effect of flanker relatedness in the main analysis was mostly driven by non-identical *related errors*. *Unrelated errors* also showed a facilitatory effect, whereas *repetition errors* showed an inhibitory effect.

To further probe the observed difference in errors rates, we calculated the odds ratio (OR), signifying how more likely it was to make an error that was affected by the flanker word (i.e., a *related error* or a *repetition error*) than an error unaffected by the flanker word (i.e., an *unrelated error* or a *no response*) separately for the related and unrelated flanker conditions and the word and bigram flanker conditions. We report the chi-squared statistic, degrees of freedom,

**Table 2. Proportion of different errors made per error category in the bigram flanker condition calculated separately for the related and unrelated flanker conditions.**

| Error Type | Flanker condition | | effect | Relatedness Effect | | |
| | Related | Unrelated | | b | SE | z |
|---|---|---|---|---|---|---|
| No response | 0.1% | 0.1% | -0.0% | -0.43 | 0.98 | -0.44 |
| Repetition error | 3.7% | 0.1% | **-3.6%** | -3.81 | 0.75 | **-5.08** |
| Related error | 2.5% | 8.3% | **5.8%** | 1.36 | 0.20 | **6.88** |
| Unrelated error | 3.3% | 6.0% | **2.7%** | 0.64 | 0.19 | **3.36** |
| Total | 9.6% | 14.5% | **4.9%** | 0.54 | 0.12 | **4.35** |

*Note.* The total error rate corresponds to the data used in the main analysis.

*p*-value, and the 95% confidence interval (CI). When flankers were full words, participants were 1.91 times (CI: 1.25; 2.92) more likely to make a *related / repetition error* in the related flanker condition compared to the unrelated flanker condition, and this difference was significant ($\chi^2$ (1) = 8.36, *p* = .004). For bigram flankers, participants were 1.36 times (CI: 0.95; 1.95) more likely to make a *related / repetition error* in the related flanker condition compared to the unrelated flanker condition, but this difference was not significant ($\chi^2$(1) = 2.46, *p* = .12).

## Discussion

Experiment 2 tested the prediction that when participants are instructed to identify central target words, rather than make lexical decisions to them (Experiment 1), then the effects of orthographically related flanker words should become inhibitory. This is precisely what was found when testing the same stimuli as Experiment 1 (word flankers) and using the same stimulus durations. On the other hand, facilitatory effects of flanker relatedness were found in the bigram flanker condition. The results of a post-hoc analysis of the different categories of error (see Tables 1 and 2) provide support for our hypothesis that there is a trade-off between inhibitory and facilitatory effects of flanker relatedness, with inhibition dominating when the flankers are whole words. We found inhibitory effects of flanker relatedness in *repetition errors* for both word and bigram flankers. Furthermore, *related errors* and *unrelated errors* showed facilitatory effects of flanker relatedness with both word and bigram flankers, although the effects were not significant when the flankers were words. Thus, the overall pattern of effects seen in Tables 1 and 2 points to the operation of both an inhibitory and a facilitatory mechanism independently of the type of flanker (word or bigram), but with inhibitory processes dominating when the flankers are words and facilitatory processes dominating when the flankers are bigrams.

## General discussion

In two online experiments using the reading version of the flankers task (Dare & Shillcock [7]), we examined the impact of orthographically related flankers on the processing of central word targets. Experiment 1 required participants to make lexical decisions on central targets, and replicated the findings of Snell et al. [10] that orthographically related flanker words (e.g., BLUE BLUR BLUE) facilitate lexical decisions to targets compared with unrelated flanker words (e.g., STEP BLUR STEP). Here these facilitatory flanker neighbor effects were observed with much shorter stimulus durations (50 ms) than used in prior research with the flankers task, and therefore provides further support as to the sublexical, and highly automatized nature of these effects. This pattern of facilitatory effects of orthographic neighbors is in line with the effects found with a parafoveal-on-foveal manipulation during sentence reading (Snell et al. [10]).

The results of Experiment 1 therefore provide further support for the overarching hypothesis guiding this and related research that the spatial integration of orthographic information operates sublexically (Grainger et al. [12]; see Fig 1). The present study more specifically aimed at examining how the subsequent activation of multiple lexical representations affects target word processing. The general hypothesis to be tested was that when participants are instructed to identify target words (Experiment 2), then inhibitory effects of flanker relatedness should be found. In the analysis of Experiment 2, this was found to be the case when flanker stimuli were whole words, but not when they were bigrams, where facilitatory effects of flanker relatedness were observed. This finding is in line with our prediction that inhibitory effects of flanker relatedness should be stronger when the flankers are whole words.

The results of a post-hoc analysis of the different types of error made by participants in Experiment 2 provide further support for the hypothesized trade-off between inhibitory and facilitatory effects of flanker relatedness that is modulated by the lexical or non-lexical nature of flanking stimuli. Indeed, the fact that the inhibitory effect of flanker relatedness in *repetition errors* was also seen in the bigram flanker condition provides support for the operation of lateral inhibition in the central processing channel of Fig 1. Related bigram flankers (e.g., BL– UE), although not presented as whole words, would activate the competing word "blue" hence causing the erroneous report of this word instead of the target "blur". Moreover, the overall facilitatory effects seen in *related and unrelated errors* for both bigram and word flankers provide strong support for the hypothesis that orthographic information is pooled across target and flanker stimuli independently of the lexical status of flankers. The overall pattern of flanker effects seen in Experiment 2 would therefore reflect a trade-off between facilitation generated by spatial pooling and inhibition generated by lexical competition. Sublexical spatial pooling would not be affected by the lexical status of flanker stimuli, whereas lexical inhibition would be greater for word flankers than bigram flankers because word flankers provide more input to competing words than do bigram flankers. The trade-off between these two influences accounts for the overall facilitatory effects of bigram flankers and inhibitory effects of word flankers.

The present findings provide further support for the interpretation of orthographic flanker effects (Dare & Shillcock [7]) proposed by Grainger et al. [12], and implemented in the OB1-reader model of text reading (Snell, van Leipsig et al. [15]). According to this account, orthographic flanker effects reflect the spatial integration of orthographic information spanning multiple spatially distinct stimuli into a single processing channel for sublexical orthographic processing and word identification. In the present study we aimed to investigate the role played by lateral inhibitory connections between co-active word representations in driving flanker effects. We hypothesized that this should depend on the task that participants are requested to perform on central target words. Our results suggest that this is indeed the case. When the task requires identification of a specific word (as opposed to lexical decision), effects of lexical competition emerge over and above the facilitatory effects driven by the sublexical spatial pooling of orthographic information across target and flankers. Crucially, and in line with this interpretation, our results show that the inhibitory influence of lexical competition has a stronger impact when flankers are whole words.

## Supporting information

**S1 Appendix. Bayesian LMM analyses of the results of Experiment 2.**
(DOCX)

## Author Contributions

**Conceptualization:** Jonathan Grainger.

**Data curation:** Aaron Vandendaele.

**Formal analysis:** Aaron Vandendaele.

**Funding acquisition:** Aaron Vandendaele, Jonathan Grainger.

**Methodology:** Jonathan Grainger.

**Supervision:** Jonathan Grainger.

**Visualization:** Aaron Vandendaele.

**Writing – original draft:** Aaron Vandendaele, Jonathan Grainger.

**Writing – review & editing:** Aaron Vandendaele, Jonathan Grainger.

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
