## [Decision Letter · Decision Letter 0]

14 Mar 2023

PONE-D-23-02898Lexical competition in the flankers task revisitedPLOS ONE

Dear Dr. Vandendaele,

Thank you for submitting your manuscript to PLOS ONE. We have received feedback from two independent reviewers, both of whom agree that the two experiments were well-designed and recommend publication with minor revisions. We invite you to submit a revised version of the manuscript that addresses the points raised by the reviewers.

The reviewers' feedback is clear and has been included below. I would like to draw your attention to a point made by Reviewer 2 regarding the random structure of the linear mixed-effects model (LMM). Instead of stating that "if the model structure allowed for it, the by-item and by-participant random random intercepts were included", it may be more direct to report which random effects were included in the final model (please also note that the word "random" was repeated, which may be a typo). Although this information is available in the online materials, providing a brief description in the main text would be helpful. 

We look forward to receiving your revised manuscript.

Kind regards,

Yiu-Kei Tsang

Academic Editor

PLOS ONE

Journal Requirements:

4. Please expand the acronym “ERC and FWO” (as indicated in your financial disclosure) so that it states the name of your funders in full.

Reviewers' comments:

Reviewer's Responses to Questions

**Comments to the Author**

1. Is the manuscript technically sound, and do the data support the conclusions?

Reviewer #1: Yes

Reviewer #2: Yes

2. Has the statistical analysis been performed appropriately and rigorously? 

Reviewer #1: Yes

Reviewer #2: I Don't Know

3. Have the authors made all data underlying the findings in their manuscript fully available?

Reviewer #1: Yes

Reviewer #2: Yes

4. Is the manuscript presented in an intelligible fashion and written in standard English?

Reviewer #1: Yes

Reviewer #2: Yes

5. Review Comments to the Author

Reviewer #1: Review of ms “Lexical competition in the flankers task revisited”

This is a nice manuscript using the reading version of the flankers task (Dare & Shillcock, 2013) that examined lexical competition. The topic is interesting, the experiments have been competently conducted and analyzed, and the manuscript is easy to read. Thus, this is a strong candidate for publication. I have several suggestions from improvement:

-It would be better to start the Introduction with a more general paragraph indicating the importance of the topic. The beginning of “In the present study we provide a further examination of the impact of orthographic relatedness across adjacent words within a sequence of words presented simultaneously and aligned horizontally” can be put in context. In this same paragraph, the parenthetical note “e.g., “The slight blue of the lights ...” => “The slight blur of the lights ...”, where the target word the word “blur” and the preview the word “blue”” seems to miss a couple of verbs (e.g. “where the target word is the word “blur” and the preview is the word “blue”).

-Related to page 5, there is evidence when manipulating a word’s N in lexical decision vs. sentence reading: the same materials produce facilitation in lexical decision and inhibition in reading (including gaze durations, see Pollatsek et al., 1999), so some rewriting may be in order.

-On page 9, I would update the reference to lme4. Also, I believe it’s more important to indicate the version of R (i.e., the core) than the version of RStudio.

-On page 11, the “1 undefined” was actually “undefined” or rather “didn’t say”?

-footnote 1: “the posterior distribution did not include 0”. I believe the authors wanted to say “the 95% credible interval of the posterior distribution did not include 0”.

-In the General Discussion, the authors may want to spell out in more detail the implications of their study for the models of reading and word recognition. The manuscript ends a bit in an abrupt manner now.

-Figure 2 does not reflect the actual timing of the flankers in the current experiments, so it needs to be slightly modified.

-Table A1/A2. The values of R need to be with two decimal points (e.g., 1.30 would reflect a bad fit among the chains and this would still be “1”).

So I am quite positive, most of the required work is on some careful rewriting and proofreading—I may have missed a few others.

Reviewer #2: This paper reports two online experiments investigating task demands on the flanker effect during word identification. Participants made lexical decisions (Experiment 1) or identified target words (Experiment 2) in the presence of flanker words/bigrams. The results showed that related flanker words facilitated lexical decisions, replicating previous research. Related flanker words, however, were associated with increased errors in the perceptual identification task, whereas related bigrams yielded facilitation. I have a few queries that I would like the authors to address before recommending this paper for publication.

- P. 5: I think it would be useful to elaborate a bit why it is argued that “the decision to move the eyes to the next word in a sequence” does not require unique word identification. The authors cite the E-Z Reader model to support this but, although that model assumes that the trigger to begin programming a saccade to the next word is the completion of a preliminary stage of word identification, it doesn’t seem accurate to imply that unique word identification does not occur in this model.

- P. 8: Please clarify if the 2025 observations per condition were based on trials before or after the data exclusion procedures reported in the following section.

- P. 9: It is stated that the LMM models only included subject and item random intercepts “if the model structure allowed for it”. Please elaborate on the model selection procedure. I would be surprised if the models failed to converge with random intercepts. Perhaps this is a typo – did you mean random slopes here?

- The Experiment 2 data is analysed separately for the two sub experiments. Given that the targets were exactly the same in the two conditions, I think an analysis that combines the data and includes flanker type as a factor (and tests interactions with flanker relatedness) would be preferable.

6. PLOS authors have the option to publish the peer review history of their article (what does this mean?). If published, this will include your full peer review and any attached files.

Reviewer #1: No

Reviewer #2: No

---

## [Author Response · Author response to Decision Letter 0]

14 Apr 2023

Reviewer #1:

Review of ms “Lexical competition in the flankers task revisited”

This is a nice manuscript using the reading version of the flankers task (Dare & Shillcock, 

2013) that examined lexical competition. The topic is interesting, the experiments have 

been competently conducted and analyzed, and the manuscript is easy to read. Thus, 

this is a strong candidate for publication. I have several suggestions from improvement:

-It would be better to start the Introduction with a more general paragraph indicating 

the importance of the topic. The beginning of “In the present study we provide a further 

examination of the impact of orthographic relatedness across adjacent words within a 

sequence of words presented simultaneously and aligned horizontally” can be put in 

context. In this same paragraph, the parenthetical note “e.g., “The slight blue of the lights 

...” => “The slight blur of the lights ...”, where the target word the word “blur” and the 

preview the word “blue”” seems to miss a couple of verbs (e.g. “where the target word is 

the word “blur” and the preview is the word “blue”).

Response: We have added a new introductory paragraph in order to provide a better context 

for the present work, and we have corrected the parenthetical note – thanks for pointing out 

this error. Moreover, when re-writing this section, we discovered that we had not done a good 

job in describing the different paradigms, so what was the initial paragraph of the manuscript 

has been re-written, and the part concerning parafoveal-previews has been simplified and 

moved to the new introductory paragraph.

-Related to page 5, there is evidence when manipulating a word’s N in lexical decision vs. 

sentence reading: the same materials produce facilitation in lexical decision and 

inhibition in reading (including gaze durations, see Pollatsek et al., 1999), so some 

rewriting may be in order.

Response: We now cite Pollatsek et al. in the new introductory paragraph, and we now refer 

to this work when discussing possible differences between lexical decision and sentence 

reading (see new footnote 1).

-On page 9, I would update the reference to lme4. Also, I believe it’s more important to 

indicate the version of R (i.e., the core) than the version of RStudio.

Response: we now cite the correct version of R. We cite the lme4 package as it is referred to 

in the repository (i.e., https://cran.r-project.org/web/packages/lme4/citation.html). We could 

not find a more recent version for the reference, but we now mention the version of lme4 

which we used for the analyses.

-On page 11, the “1 undefined” was actually “undefined” or rather “didn’t say”?

Response: This refers to the option ‘other/undefined’ given to participants when asked to

indicate their gender. We have better specified this in the revision.

-footnote 1: “the posterior distribution did not include 0”. I believe the authors wanted 

to say “the 95% credible interval of the posterior distribution did not include 0”.

Response: That is correct, we have now corrected this in the footnote.

-In the General Discussion, the authors may want to spell out in more detail the 

implications of their study for the models of reading and word recognition. The 

manuscript ends a bit in an abrupt manner now.

Response: We have added a new paragraph at the end of the General Discussion that 

summarizes the findings and points to their implications for models of word recognition and 

reading.

-Figure 2 does not reflect the actual timing of the flankers in the current experiments, so 

it needs to be slightly modified.

Response: We apologize for this error. The figure has now been modified to display the 

correct timing.

-Table A1/A2. The values of R need to be with two decimal points (e.g., 1.30 would 

reflect a bad fit among the chains and this would still be “1”).

Response: The Rhat (R^) convergence statistic is now reported up to two decimal points.

So I am quite positive, most of the required work is on some careful rewriting and 

proofreading—I may have missed a few others.

Response: Many thanks for your positive evaluation of this work. We have done the careful 

proofreading that you requested, and we hope that all is in order now.

Reviewer #2: This paper reports two online experiments investigating task demands on 

the flanker effect during word identification. Participants made lexical decisions 

(Experiment 1) or identified target words (Experiment 2) in the presence of flanker 

words/bigrams. The results showed that related flanker words facilitated lexical decisions, 

replicating previous research. Related flanker words, however, were associated with 

increased errors in the perceptual identification task, whereas related bigrams yielded 

facilitation. I have a few queries that I would like the authors to address before 

recommending this paper for publication.

- P. 5: I think it would be useful to elaborate a bit why it is argued that “the decision to 

move the eyes to the next word in a sequence” does not require unique word 

identification. The authors cite the E-Z Reader model to support this but, although that 

model assumes that the trigger to begin programming a saccade to the next word is the 

completion of a preliminary stage of word identification, it doesn’t seem accurate to 

imply that unique word identification does not occur in this model.

Response: We have now clarified this point. It is the fact that in EZ-Reader it is the 

familiarity check and not word identification that triggers the decision to move the eyes to the 

next work that is crucial for our arguments. We have clarified this point.

- P. 8: Please clarify if the 2025 observations per condition were based on trials before or 

after the data exclusion procedures reported in the following section.

Response: This number of observations per condition is before data exclusion procedures. 

We did not apply an adaptive testing procedure to guarantee enough observations per 

condition. Rather, we based ourselves on data from previous studies which used a similar 

experiment setup. We estimated a margin of 20%, meaning 2000 observations was our preset goal. After all exclusion procedures, we ended up with 1601 observations in the 

orthographically related flanker condition and 1474 in the control flanker condition. We now 

mention this in the paper.

- P. 9: It is stated that the LMM models only included subject and item random intercepts 

“if the model structure allowed for it”. Please elaborate on the model selection 

procedure. I would be surprised if the models failed to converge with random intercepts. 

Perhaps this is a typo – did you mean random slopes here?

Response: This was indeed a typo. To clarify this, we now specify the model selection 

procedure. Next to that, the final model for each analysis has been written out.

- The Experiment 2 data is analysed separately for the two sub experiments. Given that 

the targets were exactly the same in the two conditions, I think an analysis that combines 

the data and includes flanker type as a factor (and tests interactions with flanker 

relatedness) would be preferable.

Response: We now present the analysis of Experiment 2 as recommended by this reviewer. 

We obtain the same pattern of results, but with the addition of a significant interaction effect 

between type of flanker and flanker relatedness. This significant interaction allows us to 

examine, as we did before, the effects of flanker relatedness separately for the word and 

bigram flankers. We have also included a new figure (Figure 3) that provides an overview of 

these results and that highlights the significant interaction.

---

## [Editor Report · Decision Letter 1]

19 Apr 2023

Lexical competition in the flankers task revisited

PONE-D-23-02898R1

Dear Dr. Vandendaele,

We’re pleased to inform you that your manuscript has been judged scientifically suitable for publication and will be formally accepted for publication once it meets all outstanding technical requirements.

Kind regards,

Yiu-Kei Tsang

Academic Editor

PLOS ONE
---

## [Editor Report · Acceptance letter]

19 May 2023

PONE-D-23-02898R1 

Lexical competition in the flankers task revisited 

Dear Dr. Vandendaele:

I'm pleased to inform you that your manuscript has been deemed suitable for publication in PLOS ONE. Congratulations! Your manuscript is now with our production department. 

Kind regards, 

on behalf of

Dr. Yiu-Kei Tsang 

Academic Editor

PLOS ONE